# Enduring Effects of Conditional Brain Serotonin Knockdown, Followed by Recovery, on Adult Rat Neurogenesis and Behavior

**DOI:** 10.3390/cells10113240

**Published:** 2021-11-19

**Authors:** Maria Sidorova, Golo Kronenberg, Susann Matthes, Markus Petermann, Rainer Hellweg, Oksana Tuchina, Michael Bader, Natalia Alenina, Friederike Klempin

**Affiliations:** 1School of Life Sciences, Immanuel Kant Baltic Federal University, 236041 Kaliningrad, Russia; sidorova.mari@list.ru (M.S.); oktuchina@gmail.com (O.T.); 2Max Delbrück Center for Molecular Medicine, 13125 Berlin, Germany; susann.matthes.mail@gmail.com (S.M.); markus.petermann@pharmazie.uni-halle.de (M.P.); mbader@mdc-berlin.de (M.B.); alenina@mdc-berlin.de (N.A.); 3Department of Psychiatry, Psychotherapy, and Psychosomatics, Psychiatrische Universitätsklinik, 8032 Zürich, Switzerland; golo.kronenberg@pukzh.ch; 4Department of Psychiatry and Psychotherapy, Charité University Medicine, 10117 Berlin, Germany; rainer.hellweg@charite.de; 5Department of Clinical Pharmacy and Pharmacotherapy, Institute of Pharmacy, Martin Luther University, 06120 Halle, Germany; 6Institute of Translational Biomedicine, St. Petersburg State University, 199034 St. Petersburg, Russia

**Keywords:** serotonin, *Tph2*, depression, neurogenesis, stem cells, BrdU, behavior

## Abstract

Serotonin (5-hydroxytryptamine, 5-HT) is a crucial signal in the neurogenic niche of the hippocampus, where it is involved in antidepressant action. Here, we utilized a new transgenic rat model (TetO-shTPH2), where brain 5-HT levels can be acutely altered based on doxycycline (Dox)-inducible shRNA-expression. On/off stimulations of 5-HT concentrations might uniquely mirror the clinical course of major depression (e.g., relapse after discontinuation of antidepressants) in humans. Specifically, we measured 5-HT levels, and 5-HT metabolite 5-HIAA, in various brain areas following acute tryptophan hydroxylase 2 (*Tph2*) knockdown, and replenishment, and examined behavior and proliferation and survival of newly generated cells in the dentate gyrus. We found that decreased 5-HT levels in the prefrontal cortex and raphe nuclei, but not in the hippocampus of TetO-shTPH2 rats, lead to an enduring anxious phenotype. Surprisingly, the reduction in 5-HT synthesis is associated with increased numbers of BrdU-labeled cells in the dentate gyrus. At 3 weeks of *Tph2* replenishment, 5-HT levels return to baseline and survival of newly generated cells is unaffected. We speculate that the acutely induced decrease in 5-HT concentrations and increased neurogenesis might represent a compensatory mechanism.

## 1. Introduction

Pharmacotherapy for major depression frequently relies on serotonin (5-HT)-targeting medications. Chronic treatment with selective serotonin reuptake inhibitors (SSRIs) for several weeks elicits behavioral effects in tandem with distinct neuroanatomical changes, i.e., increased cell proliferation and the generation of new neurons in the dentate gyrus of experimental rodents [1,2,3]. We and others have used genetically modified mice constitutively depleted of brain 5-HT, i.e., *Tph2^−/−^*, *Tph2* knock-in, *Pet1^−/−^*, and VMAT2^SERT-Cre^ mice to study various effects of 5-HT signaling. Briefly, mice lacking brain 5-HT display transient growth retardation [4] and a “lean” phenotype [5] alongside reduced anxiety and hyperactivity/impulsivity [6,7]. In the absence of 5-HT from the adult rodent brain, survival of newly generated neurons was increased without changes in cell proliferation [8,9]. Furthermore, BDNF concentrations were enhanced in the hippocampus and prefrontal cortex of *Tph2^−/−^* mice and rats, and *Tph2* knock-in mice [10,11,12], possibly as a compensatory way of maintaining neurogenesis in the dentate gyrus. Using these mice, we found that the effects of citalopram (a commonly prescribed SSRI) on the generation of new neurons and on BDNF protein levels are largely separable [9].

A crucial limitation of the transgenic mouse models investigated so far is that one cannot firmly distinguish between phenotypes induced by the constitutive lack of 5-HT and compensatory responses provoked by its life-long absence. Therefore, we here utilize TetO-shTPH2 transgenic rats exhibiting doxycycline (Dox)-inducible expression of shRNA directed against tryptophan hydroxylase (TPH) 2, resulting in reduced *Tph2* mRNA expression and reduced brain 5-HT biosynthesis [13]. The specific advantage of TetO-shTPH2 rats lies in the fact that switch on/switch off paradigms can be used so that both the effects of 5-HT depletion as well as the enduring effects of 5-HT depletion after 5-HT levels have recovered can be investigated. Importantly, this is the first study to examine long-term consequences of transient 5-HT depletion.

In the experiments reported here, we assessed the effects of 5-HT depletion and of 5-HT depletion followed by recovery on adult neurogenesis and behavior in rats. Using Dox, we manipulated 5-HT levels in adult brain and studied, at different intervals, 5-HT turnover as well as proliferation and survival of newly generated cells in the dentate gyrus.

## 2. Materials and Methods

### 2.1. Animals and Housing Conditions

Sprague–Dawley rats (RjHAN:SD; purchased from Janvier labs, Le Genest-Saint-Isle France) were used to generate TetO-shTPH2 transgenic rats that exhibit inducible reduction of *Tph2* gene expression under Dox (doxycycline hyclate; Sigma-Aldrich Chemie GmbH, Taufkirchen, Germany) treatment [13]. When Dox treatment is stopped, *Tph2* mRNA levels recover, resulting in *Tph2* translation and 5-HT biosynthesis. Two to three rats were held in individually ventilated cages under laboratory conditions with a light/dark cycle of 12 h and free access to food and water. In line with requirements for animal keeping, cages were fitted with a reusable polycarbonate enrichment product. 

### 2.2. Experimental Design and Behavior Testing

Six-week-old female TetO-shTPH2 rats (*n* = 37) were randomly assigned to daily intraperitoneal (i.p.) injection of either Dox or 0.9% saline for 14 days. All injections were performed shortly before the active phase of the animals. In the first section of the experiment, rats received 20 mg/kg bodyweight Dox (diluted in 0.9% saline; Dox20) for 14 days and were killed on the following day, 15 (*Tph2* knockdown; Figure 1A). In the second section, animals were given a dose of 25 mg/kg bodyweight Dox for 14 days (Dox25). Half of the animals were killed on day 15 (*Tph2* knockdown), while the other half were kept for an additional 3 weeks without further treatment. These rats were killed on day 37 (*Tph2* replenishment; Figure 1A). 5-HT and 5-HIAA concentrations in the raphe nuclei (RN), prefrontal cortex (PFC), and hippocampus (HC), as well as cell proliferation and survival of newly generated cells in the dentate gyrus were determined. In addition, animals of the long-term group (Dox25 plus 3 weeks without treatment) were investigated in the Open Field test on days 15 and 35 to determine overall activity/locomotion and anxiety-like behavior. Briefly, rats were individually placed into an open arena measuring 86 cm × 86 cm × 40 cm (made of white non-odor-absorbent material, the center of the arena comprises 50%) under dim lighting conditions. Activity monitoring per rat was conducted for 5 min and shortly before the active phase began. The resulting data were analyzed using BehaviorCloud LLC.2 (Columbus, OH, USA).

### 2.3. Thymidine Analog Injections

Cell proliferation was investigated at 15 days following daily drug injections, and cell proliferation and survival at 37 days when Dox had been withdrawn for 3 weeks. Two thymidine analog injection paradigms were used to determine both the effect of inducible *Tph2* knockdown and subsequent replenishment on progenitor cells. To analyze cell proliferation, rats received three i.p. injections, 5 h apart, of BrdU (5-bromo-2′-deoxyuridine, 50 mg/kg bodyweight; Sigma-Aldrich Chemie GmbH, Taufkirchen, Germany) on day 14 following daily Dox or saline treatment, and were killed 24 h after the first BrdU injection. Cell survival and long-term cell proliferation over 3 weeks during *Tph2* replenishment were determined using the combination of two other halogenated thymidine analogs, CldU (5-chloro-2′-deoxyuridine) and IdU (5-iodo-2′-deoxyuridine) [14]. First, on day 14, animals received three i.p. injections, 5 h apart, of CldU (42.5 mg/kg bodyweight; Sigma-Aldrich Chemie GmbH, Taufkirchen, Germany) and rats were left 3 weeks to assess survival of the labeled cohort. One day prior to tissue collection at 37 days, IdU (57.5 mg/kg bodyweight diluted in 1 mL 0.2 N NaOH; Sigma-Aldrich Chemie GmbH, Taufkirchen, Germany) was administered three times, 5 h apart, to label proliferating cells.

### 2.4. Tissue Preparation and Procedures

Rats were deeply anesthetized and perfused transcardially with 0.9% saline. Brains were removed and hemispheres were separated. The right hemisphere was placed into 4% paraformaldehyde overnight (followed by 30% sucrose) for immunohistochemistry. The dissected HC and PFC of the left hemisphere and the raphe region from both hemispheres were snap-frozen for high-performance liquid chromatography (HPLC) analyses. Briefly, endogenous levels of 5-HT and its metabolite, 5-HIAA, were measured from thawed homogenates (homogenized in lysis buffer containing ascorbic and perchloric acid) using high-sensitive HPLC with fluorometric detection (Shimadzu, Tokyo, Japan). Raw data were normalized to wet tissue weight, and results to saline-treated controls (100%) for statistical analysis. Serotonin turnover is depicted as 5-HIAA/5-HT ratio.

### 2.5. Immunohistochemistry and Quantification

Immunohistochemistry followed the peroxidase method in accordance with an established protocol [15]. Briefly, one-in-eight series of sequential 50 µm coronal brain sections were stained free floating. For thymidine labeling, DNA was denatured in 2 N HCl for 20 min at 37 °C. Sections were rinsed in 0.1 M Borate buffer and washed in Tris-buffered saline (TBS); all antibodies were diluted in TBS containing 3% donkey serum and 0.1% Triton X-100. BrdU-, CldU-, or IdU-positive cells were counted throughout the rostro-caudal extent of the dentate gyrus. The total number of labeled cells was estimated by multiplying cell counts by eight. For immunofluorescence, 50 to 70 BrdU- and IdU-positive cells per dentate gyrus were randomly selected for phenotypic analysis using a Leica TCS SP5 confocal microscope (Leica, Wetzlar, Germany). Primary antibodies were applied in the following concentrations: anti-BrdU (rat, 1:500; AbD serotec), anti-CldU (rat anti-BrdU, 1:500; AbD serotec), anti-doublecortin (DCX; goat, 1:250; Santa Cruz Biotechnology, Heidelberg, Germany), anti-IdU (mouse, clone 32D8.D9, 1:1000; Biomol, Hamburg, Germany), and anti-Sox2 (goat, 1:1000; Santa Cruz Biotechnology). For immunofluorescence, secondary antibodies conjugated to Alexa488, Cy3, and Cy5 were used (1:250, Invitrogen, ThermoFisher, Waltham, MA USA).

Statistical analysis. One-way ANOVA was followed by Dunnett’s or Tukey’s post hoc test (GraphPad Prism software v9.0.2 San Diego, CA, USA). Student’s *t* test was used for individual pairwise comparisons. All values are expressed as mean ± SEM. *p* values of ≤0.05 were considered statistically significant.

## 3. Results

### 3.1. Acute Knockdown of Tph2 Expression Decreases 5-HT Levels in the PFC and RN

We used TetO-shTPH2 transgenic rats to manipulate central 5-HT concentrations under Dox treatment. Animals were given a daily i.p. dose of either 20 or 25 mg/kg bodyweight Dox for 14 days, and we assessed 5-HT levels in several brain areas by HPLC. Fourteen days of continued i.p. administration of Dox to TetO-shTPH2 rats significantly decreased 5-HT levels in the PFC to 70.9 ± 5.9% with Dox20 and 58.4 ± 5.7% with Dox25 of values in saline-treated control rats (100%; F(2,21) = 13.99, *p* = 0.0001, Dunnett’s post hoc test *p_Dox20_* = 0.024, *p_Dox25_* = 0.0002; Figure 1B). In the RN, 5-HT levels were reduced to 78.4 ± 12.4% (Dox20) and 45.3 ± 4.7% (Dox25) of the control value (F(2,18) = 5.227, *p* = 0.0162, Dunnett’s post hoc test *p_Dox20_* = 0.2563, *p_Dox25_* = 0.0096; Figure 1B). However, no significant decrease in 5-HT levels in the HC of TetO-shTPH2 rats was observed for either Dox dosage (Dox20, 79.4 ± 11.1% and Dox25, 89.6 ± 7.3% of the control value; F(2,18) = 1.660, *p* = 0.2142; Figure 1B). Overall, the results with Dox25 show a successful reduction in brain 5-HT levels in PFC and RN, which exceeds the reduction that is achieved by oral administration [13].

### 3.2. Retrieval of Tph2 Expression Results in the Recovery of 5-HT Levels

In an effort to mimic antidepressant effects in depressed patients, namely increasing 5-HT levels, we discontinued Dox25 administrations so as to allow renewed 5-HT synthesis. At 3 weeks without treatment, 5-HT levels had recovered in the RN to 80.1 ± 13.4% (F(2,15) = 0.40, *p* = 0.679; Figure 1C), and in the PFC to 70.6 ± 7.3% of the control value (100%; F(2,14) = 1.38, *p* = 0.284; Figure 1C). Serotonin levels in HC were unaffected (82.1 ± 4.9%, F(2,15) = 1.32, *p* = 0.298). Furthermore, the reduction in 5-HT concentrations following *Tph2* knockdown with Dox25 translated into reduced 5-HIAA levels and 5-HIAA/5-HT turnover rate in RN (*p* = 0.027) and PFC (*p* = 0.019; Figure 1D). Notably, 5-HT usage was also significantly affected in HC upon *Tph2* knockdown (*p* = 0.007; Figure 1D). No significant differences in 5-HIAA/5-HT turnover rates were detected between groups following *Tph2* replenishment (Figure 1D). Together, 5-HT synthesis was restored 3 weeks following its conditional inhibition.

### 3.3. Acute Knockdown of Tph2 Expression Increases Cell Proliferation in the Dentate Gyrus

Next, we assessed whether downregulation of *Tph2* expression affects precursor cell proliferation in the HC (Figure 1E). Remarkably, 14 days of acute manipulation of 5-HT levels in the adult brain of TetO-shTPH2 rats significantly increased the number of BrdU-labeled cells in the dentate gyrus (Saline 888 ± 142 cells vs. Dox20 1597 ± 229 cells, *p* = 0.0212; saline 823 ± 43 cells vs. Dox25 1303 ± 89 cells, *p* = 0.0006; Figure 1F). We probed BrdU-positive cells for co-expression of the lineage markers Sox2 (precursor cells) and DCX (transient immature neurons). Phenotypic analysis of the Dox25 group revealed an increase in the number of BrdU/Sox2 (saline 227 ± 20 vs. Dox25 331 ± 39 cells, *p* = 0.0380) and BrdU/DCX-expressing cells (saline 312 ± 8 vs. Dox25 457 ± 56 cells, *p* = 0.0278; Table 1) compared to saline-treated control rats. These results may reveal a compensatory effect on cell proliferation following acute depletion of brain 5-HT concentrations.

### 3.4. Enduring Effects of 5-HT Depletion on Cell Proliferation in the Dentate Gyrus

Two other halogenated thymidine markers were used to determine survival (CldU) and long-term proliferation (IdU) of these newly generated cells after 5-HT depletion was stopped followed by recovery. The stimulation of progenitor cells (BrdU-labeled) observed following *Tph2* knockdown did not lead to increased survival (CldU-labeled) at 3 weeks of *Tph2* replenishment—instead, no difference in cell numbers was detected between saline and Dox25 (CldU, saline 832 ± 62 cells vs. Dox25 902 ± 125 cells, *p* = 0.935; Figure 1G). However, 5-HT depletion followed by recovery significantly reduced the number of proliferating cells at day 37, labeled with IdU at day 36 (saline 1393 ± 99 cells vs. Dox25 981 ± 87 cells, *p* = 0.0109; Figure 1G). Although numbers were reduced, phenotypic analysis of IdU-positive cells revealed a similar shift in the distribution of Sox2 and DCX-co-expressing cells that has been determined for BrdU (IdU/Sox2, saline 24.6 ± 1.8 vs. Dox25 30.3 ± 1.6%, *p* = 0.0379; IdU/DCX, saline 47.1 ± 2.9 vs. Dox25 58.0 ± 2.2%, *p* = 0.0158; Table 1). Notably, we observed a positive correlation between CldU and IdU cell numbers at Dox25 (CldU/IdU R^2^ = 0.6859, *p* = 0.0417; Figure 1H) suggesting surviving cells remain a proliferative capacity, which was absent in saline-treated control rats. Representative light microscopy images show the differences in cell numbers for proliferation, cell survival and long-term cell proliferation between saline and Dox25 (Figure 1I). These data suggest long-term consequences of acute 5-HT depletion on cell proliferation in the dentate gyrus.

**Table 1 cells-10-03240-t001:** Number and phenotypes of BrdU- and IdU-positive cells in the SGZ/GCL. BrdU- and IdU-positive cells co-expressing Sox2 (precursor cells, type-2) or DCX (transient immature neurons, type 2b/3). All data are shown as mean ± sem. GCL, granule cell layer, SGZ, subgranular zone, (#) numbers, (%) percentage.

**Short-Term**	**BrdU Numbers**	**BrdU/Sox2**	**BrdU/DCX**
Control (#)	823 (43)	227 (20)	312 (8)
Dox25 (#)	1303 (89) ***	331 (39) *	457 (56) *
Control (%)		27.8 (2.4)	38.4 (2.2)
Dox25 (%)		25.3 (1.8)	34.6 (2.5)
**Long-Term**	**IdU Numbers**	**IdU/Sox2**	**IdU/DCX**
Control (#)	1393 (99)	331 (45)	681 (67)
Dox25 (#)	981 (87) **	296 (25)	577 (77)
Control (%)		24.6 (1.8)	47.1 (2.9)
Dox25 (%)		30.3 (1.6) *	58.0 (2.2) *

*** *p* ≤ 0.001, ** *p* ≤ 0.01, * *p* ≤ 0.05 to control.

### 3.5. Acute Knockdown of Tph2 Expression Leads to a More Anxious, Enduring Phenotype

We subjected animals of the long-term/recovery group to the Open Field test (Figure 1J) to assess whether manipulations in 5-HT levels lead to alterations in locomotor activity based on motivation and anxiety. At 2 weeks of *Tph2* knockdown, rats spent significantly less time in the center of the Open Field arena, a behavior which was not reversed by *Tph2* replenishment at 35 days (in %, F(3,20) = 5.694 *p* = 0.0055; Tukey’s post hoc test *p*_day15_ = 0.0112, *p_day35_* = 0.020; Figure 1K). Start latency entering the center was accordingly increased at both time points (in s, F(3,20) = 4.789 *p* = 0.0113; *p_day15_* = 0.0312, *p_day35_* = 0.0186; Figure 1K). However, activity levels of TetO-shTPH2 rats upon *Tph2* knockdown and *Tph2* replenishment were similar to saline control as shown by activity distribution (F(3,20) = 1.662, *p* = 0.2072; Figure 1L) and average velocity (F(3,20) = 1.987, *p* = 0.1484; Figure 1L). Less time spent in centers while overall locomotion was unchanged suggests that rats are more anxious when 5-HT levels were transiently reduced.

## 4. Discussion

Our data demonstrate the acute and enduring effects of transitory 5-HT depletion on adult neurogenesis and behavior in transgenic rats. We found that acutely lowered brain 5-HT leads to a transient increase in the number of newborn cells in the dentate gyrus; accompanied by a more anxious behavioral phenotype. We show the successful recovery of 5-HT concentrations in RN and PFC at 3 weeks following the induced *Tph2* knockdown. Although 5-HT levels returned to normal, survival of newly generated cells and anxious behavior remained unaffected resulting in enduring neuroanatomical and behavioral consequences of 5-HT diminution.

The genetic approach used here allowed the (i) on/off stimulation of 5-HT levels within the same animal, and (ii) dissecting of pure 5-HT-dependent effects on neuroplasticity and behavior—an advantage over previous studies in rodent models of permanent 5-HT depletion [16]. Our main result of increased cell proliferation following reduced 5-HT synthesis is, for the moment, counterintuitive considering that the believed mechanism of SSRI action is increased availability of 5-HT accompanied by increased neurogenesis [2,3]. At 3 weeks of 5-HT recovery, however, no increase in cell survival and neurogenesis was observed. Instead, the correlation of CldU/IdU-labeling suggests that surviving cells maintain proliferative capacity and no further recruitment of progenitor cells occurs (decreased IdU-only expressing cells). Enhanced proliferation following the induced *Tph2* knockdown may have transiently exhausted the neural stem/progenitor cell pool. Phenotypic analysis reveals advanced lineage progression to maintain neurogenesis. An enduring effect was also observed in the prolonged anxiety-like behavior in the Open Field. We speculate that refilled 5-HT levels only affect behavior at a later time point, which would also require increased cell survival.

The neurogenic niche of the HC is main target for dense 5-HT fiber projections from RN [17]. Although high-dosages of Dox induced a significant reduction in 5-HT levels in RN and PFC, levels remained high in HC; yet, with a reduced turnover rate. Different explanations are possible: (i) excessive storage of 5-HT at synapses/boutons *en passant*, (ii) a slowed release of the neurotransmitter in the acute reduction of 5-HT synthesis, or (iii) increased 5-HT receptor expression on granule and progenitor cells to maintain homeostasis. Serotonin fibers have been shown to remodel and transform in response to alterations in 5-HT [18]. The acute decrease in brain 5-HT levels in our study might have induced 5-HT fiber plasticity, leading to increased cell proliferation. In addition, the PFC is a newly identified region involved in 5-HT mechanisms [19] aiming at hippocampal function [20]. Thus, lowered 5-HT levels found in PFC might in turn affect neuroplasticity in the dentate gyrus. This interesting interconnectivity needs further investigation.

In this study, we manipulated 5-HT synthesis in the adult rat brain to uniquely reflect the episodic course of depression in humans. We examined neurobiology resulting from alternating 5-HT supply and observed enduring mechanisms in the acute event of 5-HT manipulation. The results might impact antidepressant treatment regimes in patients.

## Figures and Tables

**Figure 1 cells-10-03240-f001:**
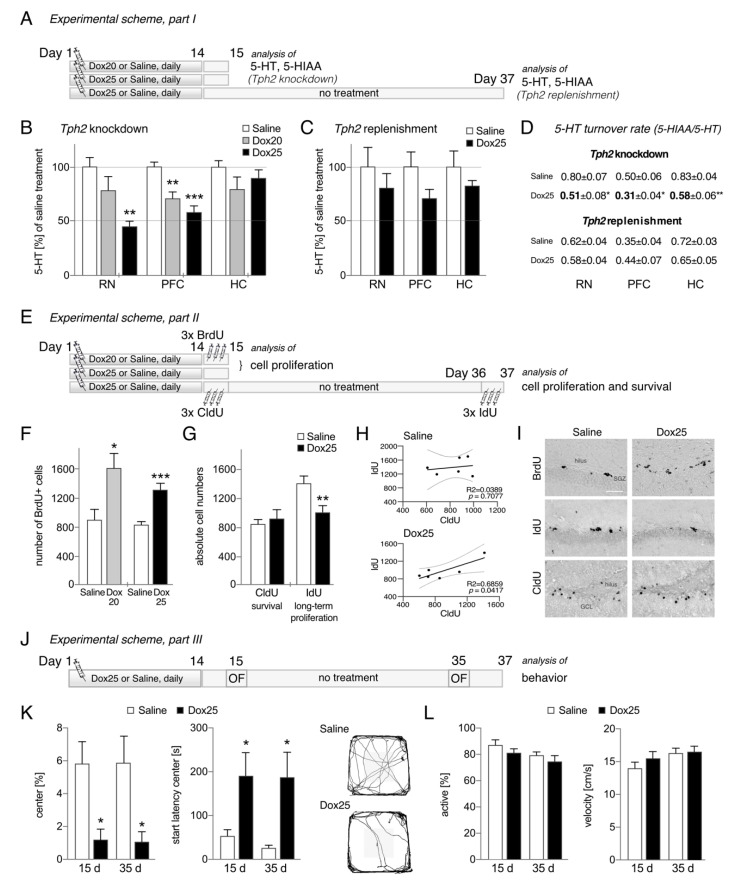
Effects of conditional brain serotonin knockdown, followed by recovery. (**A**) Experimental design part I: *Tph2* knockdown and *Tph2* replenishment to manipulate serotonin (5-HT) levels. TetO-shTPH2 transgenic rats received daily treatment of either saline, Dox20, or Dox25 for 14 days to determine 5-HT concentrations and its metabolite, 5-HIAA, in brain tissue on day 15 following *Tph2* knockdown. Another group of TetO-shTPH2 rats receiving either saline or Dox25 was held for an additional 3 weeks (no treatment, *Tph2* replenishment) to analyze 5-HT and 5-HIAA levels at day 37. (**B**) Fourteen days of continued i.p. Dox administration into TetO-shTPH2 rats significantly decreased 5-HT levels in RN (upon Dox25) and PFC (upon Dox20 and Dox25) with no changes observed in the HC. Dunnett’s post hoc tests ** *p* < 0.01, *** *p* < 0.001. (**C**) Following 3 weeks without treatment (*Tph2* replenishment), 5-HT levels had recovered in the RN and PFC; no differences between treatment groups were observed. (**D**) A significant decrease in the rate of 5-HT usage in RN, PFC, and HC was determined following *Tph2* knockdown while the differences were obsolete following *Tph2* replenishment; Student’s *t* test * *p* < 0.05, ** *p* < 0.01; RN, raphe nuclei, PFC, prefrontal cortex, HC, hippocampus. (**E**) Experimental design part II: Cell proliferation and survival in the dentate gyrus. At 14 days following treatment with saline, Dox20, or Dox25, a first group of TetO-shTPH2 rats received three injections of BrdU and were killed the following day to analyze cell proliferation. A second group (saline- and Dox25-treated) received two other halogenated thymidine markers for long-term analysis: CldU was injected three times on day 14 and cell survival was analyzed at day 37; IdU was injected on day 36 and rats were killed the next day. (**F**) Fourteen days of continued i.p. administration of Dox significantly increased the number of BrdU-positive cells in the dentate gyrus of Dox20 and Dox25 groups compared with saline. Student’s *t* tests * *p* < 0.05, *** *p* < 0.001. (**G**) No difference in cell survival (CldU) was observed compared between treatment groups; while *Tph2* replenishment affects long-term cell proliferation: the number of IdU-positive cells was significantly decreased, Student’s *t* tests ** *p* < 0.01 to IdU saline control. (**H**) A positive correlation of CldU/IdU labeling was observed for the Dox25 group but was absent in saline control. (**I**) Peroxidase staining to characterize BrdU-, IdU- and CldU-labeled cells in hippocampus. BrdU and IdU injections 1 day prior to tissue collection shows the characteristic clustered distribution of proliferating cells within the SGZ that is in opposite between treatment groups, BrdU vs. IdU; CldU injected 3 weeks prior to tissue collection permits assessment of survival and reveals migration of labeled cells into the GCL. Scale bar 100 µm. BrdU, bromodeoxyuridine, CldU, 5-chloro-2′deoxyuridine, GCL, granule cell layer, IdU, 5-iodo-2′deoxyuridine, SGZ, subgranular zone. (**J**) Experimental design part III: Behavior response to *Tph2* knockdown vs. *Tph2* replenishment. Animals of the long-term group were subjected to an Open Field arena on days 15 and 35. (**K**) Dox25-treated TetO-shTPH2 rats spent significantly less time in the center of the arena (in percentage) that is accompanied by increased start latency to enter (in time) at both time points, days 15 and 35, compared with saline control, Tukey’s post hoc test * *p* < 0.05; center vs. periphery is 50%. (**L**) Locomotor behavior was intact; no differences were observed in activity levels (in percentage), and mean velocity (cm/s).

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
