# Peer review of "Enduring Effects of Conditional Brain Serotonin Knockdown, Followed by Recovery, on Adult Rat Neurogenesis and Behavior"

_cells, 2021, doi:10.3390/cells10113240_

Round 1

Reviewer 1 Report

The manuscript contains new and very important information about the “acute” and enduring effects of conditional knockdown of Tph2 on the anxiety-related behavior, 5-HT metabolism and neurogenesis in the brain of TetO-shTPH2 rats. The authors showed that the knockdown markedly reduced 5-HT level in midbrain and frontal cortex as well as 5-HT metabolism in midbrain, frontal cortex and hippocampus. These alterations in the brain 5-HT system were accompanied with marked activation of neurogenesis in hippocampus and increase of anxiety in the OF. Three weeks after the knockdown end when 5-HT metabolism was normalized, the level of anxiety still remained high.

The manuscript is well written and does not need any considerable correction before publication. However, I’d like to recommend 3 minor corrections which I think could be improve understanding the results.

1. The authors should indicate the SIZE of the arena’s center (in % of the arena). This note in important. Others two notes are optional

2. The phrase “In the first section of the experiment, rats received 20 mg/kg bodyweight Dox (diluted in 0.9% saline; Dox20) and were killed on the following day, 15 (Tph2 knockdown; Figure 1A)” (lines 77-79) should be better replaced by “In the first section of the experiment, rats received 20 mg/kg bodyweight Dox (diluted in 0.9% saline; Dox20) for 14 days and were killed on the day 15 (Tph2 knockdown; Figure 1A)”.

3. I think that term “acute” is not good for 14 days treatment (“prolonged” seems will be better).

Author Response

We thank the reviewers for their comments and have added a few word groups in response, and to clarify our original submission. Relevant changes in the manuscript have been marked using ‘Track Changes’, and a detailed response to reviewer comments follows.

Reviewer #1:

The manuscript is well written and does not need any considerable correction before publication. However, I’d like to recommend 3 minor corrections which I think could be improve understanding the results.

  1. The authors should indicate the SIZE of the arena’s center (in % of the arena). This note in important. Others two notes are optional.

We thank the reviewer for the comment, and have added the percentage center vs periphery into the text (Materials and Methods section, Discussion) as well as have drawn a small square into Figure 1L.

In M&M it now reads:

“Briefly, rats were individually placed into an open arena measuring 86x86x40 cm (made of white non-odor-absorbent material, the center of the arena comprises 50%) under dim lighting conditions.”

  1. The phrase “In the first section of the experiment, rats received 20 mg/kg bodyweight Dox (diluted in 0.9% saline; Dox20) and were killed on the following day, 15 (Tph2 knockdown; Figure 1A)” (lines 77-79) should be better replaced by “In the first section of the experiment, rats received 20 mg/kg bodyweight Dox (diluted in 0.9% saline; Dox20) for 14 days and were killed on the day 15 (Tph2 knockdown; Figure 1A)”

Although, it says ’14 days’ twice in the sentences around, we have added it for this sentence, too. It now reads:

“Six-week-old female TetO-shTPH2 rats (n = 37) were randomly assigned to daily intraperitoneal (i.p.) injection of either Dox or 0.9% saline for 14 days. All injections were performed shortly before the active phase of the animals. In the first section of the experiment, rats received 20 mg/kg bodyweight Dox (diluted in 0.9% saline; Dox20) for 14 days and were killed on the following day, 15 (Tph2 knockdown; Figure 1A). In the second section, animals were given a dose of 25 mg/kg bodyweight Dox for 14 days (Dox25).”

  1. I think that term “acute” is not good for 14 days treatment (“prolonged” seems will be better)

We thank the reviewer for this comment. However, we will keep ‘acute’ since 1st , we don’t know when exactly the 5ht levels decreased, and 2nd, wanted to put it into comparison to our earlier studies on rodents w/ life-long absence of serotonin, or treatment with antidepressants.

Reviewer 2 Report

Manuscript cells-1448866-v1 entitled "Enduring effects of conditional brain serotonin knockdown, followed by recovery, on adult rat neurogenesis and behavior" by Sidorova et al.

In this manuscript, the authors were interested in testing a new transgenic rat model (TetO-shTPH2), which they previously produced and characterized. In this transgenic rat, brain 5-HT levels can be acutely reduced by shRNA against TpH2 upon doxycycline (Dox) injection or restored after stopping Dox injection and thus modeling relapse after discontinuation of antidepressants. In this work, the authors report that Dox induces decreased 5-HT levels in the prefrontal cortex and raphe nuclei, but not in the hippocampus and is associated to anxious phenotype. Furthermore, reduced 5-HT synthesis is associated with increased proliferation in hippocampus dentate gyrus as assessed by reduction of BrdU-labeled cells. Following 3 weeks of Tph2 restoration, 5-HT levels return to baseline and survival of newly generated cells is unaffected.

This rather small set of experimental data are sound and interesting for the readers.

In this piece of work, the authors measured basal levels of 5-HT, which are not significantly reduced in hippocampus although DG proliferation is increased. This finding is puzzling knowing that they observed Reduced 5-HT in hippocampus of the same rat strain in their previous paper (Matthes et al. 2019). One would like to have deeper investigation of this finding. Is it related to Dox dosing? To uneven knockdown of Tph2 in some raphe nuclei? or else.

Author Response

We thank the reviewers for their comments and have added a few word groups in response, and to clarify our original submission. Relevant changes in the manuscript have been marked using ‘Track Changes’, and a detailed response to reviewer comments follows.

Reviewer #2: This rather small set of experimental data are sound and interesting for the readers.

In this piece of work, the authors measured basal levels of 5-HT, which are not significantly reduced in hippocampus although DG proliferation is increased. This finding is puzzling knowing that they observed Reduced 5-HT in hippocampus of the same rat strain in their previous paper (Matthes et al. 2019). One would like to have deeper investigation of this finding. Is it related to Dox dosing? To uneven knockdown of Tph2 in some raphe nuclei? or else. 

We thank the reviewer for the comment. The experiments for HPLC analysis have been done using the same strain of rats; yet, the animals in the current study were young-adult females vs adult male and females used in Matthes et al 2019. While 5ht levels were significantly reduced by 19% in male, they were only reduced by 9% (p=0.09) in female HC of adult rats. For the current study, we have used intraperitoneal injections over drinking water since it is more precise in terms of concentrations; and had Dox amounts increased. However, 5ht levels in HC remain unaffected and we suggest it is due to high fiber tracts and not RN diversity. We prefer leaving a sex-specific discussion of 5ht levels in HC out of the current study, and since it had already been discussed earlier.